# Meta-imputation of transcriptome from genotypes across multiple datasets by leveraging publicly available summary-level data

Andrew E. Liu *, Hyun Min Kang *

Department of Biostatistics and Center for Statistical Genetics, University of Michigan, Ann Arbor, Michigan, United States of America

* aeyliu@umich.edu (AEL); hmkang@umich.edu (HMK)

**Data Availability Statement:** Software and raw data files are held in: https://github.com/aeyliu/SWAM Additional scripts can be found in: https://github.com/aeyliu/SWAM-manuscript.

## Abstract

Transcriptome wide association studies (TWAS) can be used as a powerful method to identify and interpret the underlying biological mechanisms behind GWAS by mapping gene expression levels with phenotypes. In TWAS, gene expression is often imputed from individual-level genotypes of regulatory variants identified from external resources, such as Genotype-Tissue Expression (GTEx) Project. In this setting, a straightforward approach to impute expression levels of a specific tissue is to use the model trained from the same tissue type. When multiple tissues are available for the same subjects, it has been demonstrated that training imputation models from multiple tissue types improves the accuracy because of shared eQTLs between the tissues and increase in effective sample size. However, existing joint-tissue methods require access of genotype and expression data across all tissues. Moreover, they cannot leverage the abundance of various expression datasets across various tissues for non-overlapping individuals. Here, we explore the optimal way to combine imputed levels across training models from multiple tissues and datasets in a flexible manner using summary-level data. Our proposed method (SWAM) combines arbitrary number of transcriptome imputation models to linearly optimize the imputation accuracy given a target tissue. By integrating models across tissues and/or individuals, SWAM can improve the accuracy of transcriptome imputation or to improve power to TWAS while only requiring individual-level data from a single reference cohort. To evaluate the accuracy of SWAM, we combined 49 tissue-specific gene expression imputation models from the GTEx Project as well as from a large eQTL study of Depression Susceptibility Genes and Networks (DGN) Project and tested imputation accuracy in GEUVADIS lymphoblastoid cell lines samples. We also extend our meta-imputation method to meta-TWAS to leverage multiple tissues in TWAS analysis with summary-level statistics. Our results capitalize on the importance of integrating multiple tissues to unravel regulatory impacts of genetic variants on complex traits.

**Funding:** This work was supported by NIH grants HL137182 (from NHLBI, https://www.nhlbi.nih. gov/, to A.E.L and H.M.K), HG009976 (from NHGRI, https://www.genome.gov/, to H.M.K), DK082841 (from NIDDK https://www.niddk.nih. gov/, to A,E.L and H.M.K), and DK081943 (from NIDDK https://www.niddk.nih.gov/, to A,E.L and H. M.K). The authors receiving funding were A.E.L. and H.M.K. The funders had no role in study design, data collection and analysis, decision to publish, or preparation of the manuscript.

**Competing interests:** H.M.K. is presently an employee of Regeneron Pharmaceuticals, in which he owns stock and stock options. A.E.L. is presently an employee of Gencove Inc., in which he owns stock and stock options. Regeneron Pharmaceuticals and Gencove Inc. did not play a role in the study design, data collection and analysis, decision to publish, or preparation of the manuscript. There are no patents, products in development or marketed products associated with this research to declare. This does not alter our adherence to PLOS policies on sharing data and materials.

## Author summary

The gene expression levels within a cell are affected by various factors, including DNA variation, cell type, cellular microenvironment, disease status, and other environmental factors surrounding the individual. The genetic component of gene expression is known to explain a substantial fraction of transcriptional variation among individuals and can be imputed from genotypes in a tissue-specific manner, by training from population-scale transcriptomic profiles designed to identify expression quantitative loci (eQTLs). Imputing gene expression levels is shown to help understand the genetic basis of human disease through Transcriptome-wide association analysis (TWAS) and Mendelian Randomization (MR). However, it has been unclear how to integrate multiple imputation models trained from individual datasets to maximize their accuracy without having to access individual genotypes and expression levels that are often protected for privacy concerns. We developed *SWAM* (Smartly Weighted Averaging across Multiple datasets), a *meta-imputation* framework which can accurately impute gene expression levels from genotypes by integrating multiple imputation models without requiring individual-level data. Our method examines the similarity or differences between resources and borrowing information most relevant to the tissue of interest. We demonstrate that SWAM outperforms existing single-tissue and multi-tissue imputation models and continue to increase accuracy when integrating additional imputation models.

## Introduction

Genome wide association studies (GWAS) have been able to identify numerous associations between genetic variants and complex traits. However, interpreting the biological mechanisms underlying the association signals remains a challenge [1]. Recently, studies involving gene expression have become increasingly popular as a means to provide biologically meaningful insight into statistical associations [2,3]. Transcriptome-wide association studies (TWAS) is a widely used method to translate GWAS association signals into more interpretable units by examining the association between phenotypes and gene expression levels imputed from genotypes. Associations identified from TWAS can be interpreted as potentially causal relationships between the traits and the genes through gene regulation [4–6]. While TWAS may not detect associations driven by functional mechanisms irrelevant to gene regulation, it increases the specificity and interpretability in identifying GWAS signals driven by gene regulation. Imputed gene expression can be utilized in various contexts of association analysis beyond TWAS, such as Mendelian randomization [7,8] or estimation of trait heritability attributable to cis-eQTLs [9]. Since genotype data from DNA is far easier and cheaper to obtain than expression data from tissues, TWAS based on imputed expression offers excellent augmentation to study the genetic component of gene regulation in addition to RNA-seq-based studies.

The first-generation methods to impute gene expression levels from genotypes train the model from a single-tissue dataset comprising of many individuals with both genotypes and expression profiles [2,3]. For example, a widely-used method PrediXcan [2] uses Elastic net regularization to identify cis-eQTLs (expression quantitative loci) to train the model to impute gene expressions from genotypes. Other methods, such as TWAS [3], employ different regularization but typically produces a linear model to impute gene expressions as a weighted sum of cis-eQTL genotypes. Imputation models are trained using these methods from various population-scale transcriptomic datasets, such as the Genotype-Tissue Expression (GTEx) project [9,10], Depression Genes and Network (DGN) study [11], and The Cancer Genome Atlas

(TCGA) [12], and these models are made available in public repository such as predictDB (http://predictdb.org/) or FUSION (http://gusevlab.org/projects/fusion/) so that expression imputation or TWAS can be performed from any genotyped individuals.

Although these first-generation methods for transcriptome imputation have been quite useful, they have limited accuracy mostly due to limited sample size in the training datasets where both genome-wide genotypes and transcriptome-wide expression levels are available. While millions of individuals have been genotyped or sequenced to date [13–16], the sample-size of current population-scale transcriptome data are typically limited only to hundreds or thousands [17] (with the largest study cohort having around 30k participants [18]), primarily due to the difficulty in collecting high quality tissues (other than whole blood) from living donors. Moreover, transcriptomic datasets are prone to potential batch effects between studies [19–22], making it difficult to integrate across multiple datasets to build a large and harmonized resource to be trained from. Furthermore, there are hundreds or thousands of different types of tissues or cells, requiring orders of magnitude larger effort to comprehensively profile transcriptomes in population-scale across tissues, as in GTEx Project.

Recently, methods to address the shortcomings of the first-generation methods have been developed. When transcriptomic profiles are available across many tissues, such as in the GTEx Project, transcriptome imputation can improve by leveraging the shared genetic components across tissues. Even though each tissue represents a unique transcriptomic profile, a large fraction of eQTLs are shared across tissues [23], and the availability of multiple expression measurements across tissues can help more precisely identify the shared eQTLs, which in turn can improve the imputation accuracy. For example, UTMOST trains a transcriptome imputation model simultaneously across all tissues using a combination of L1 and L2 penalization across markers and tissues, respectively [24]. Another multi-tissue approach, MultiXcan, does not impute transcriptomes, but performs a multi-tissue TWAS across all tissues by using already-imputed tissue-specific expression as a predictor variable to improve power to identify association between a trait and a gene, in which the underlying mechanism potentially involves multiple tissues or cell types [25].

Even though UTMOST substantially improves the accuracy of transcriptome imputation, it assumes that expression measurements across multiple tissues are available for overlapping set of genotypes individuals for training imputation models. While this assumption can be met when training from the GTEx dataset (assuming granted access to the individual-level data), it may not be realistic in other circumstances where expression measurements are available for non-overlapping individuals (such as in TCGA), or it is infeasible to obtain individual-level genotypes and expression data due to limited access privilege. As population-scale transcriptomic resources are rapidly increasing, it should be possible in principle to integrate these resources to better impute transcriptomes. While there have been additional methods which have been developed to increase the accuracy of gene expression or TWAS [25–28], none of them–to the best of our knowledge–are able to perform "meta-imputation", which systematically integrates multiple imputation models without the need to access to individual-level data.

Here we propose Smartly Weighted Averaging across Multiple datasets (SWAM), a multi-tissue transcriptome imputation method based on a flexible meta-analysis across multiple imputation models. Unlike UTMOST, SWAM does not require access to all genotypes and expression datasets for training its imputation model. Instead, it takes individual transcriptome imputation models trained from individual tissues while optimizing the expected imputation accuracy for a target tissue. Moreover, it can seamlessly integrate imputation models trained from multiple datasets comprising of different individuals and tissues. As a result, SWAM can integrate across hundreds of imputation models across GTEx, DGN, and TCGA projects without requiring all individual-level data to substantially improve the imputation

accuracy over existing methods, as we demonstrate with GEUVADIS data. Moreover, we demonstrate that SWAM improves the power of TWAS over single-tissue methods and many alternative multi-tissue methods.

## Results

### Smartly Weighted Averaging across Multiple Datasets (SWAM)

We propose *Smartly Weighted Averaging across Multiple datasets* (SWAM), a method that provides a flexible framework to impute tissue-specific expression by integrating single-tissue imputation models derived from other tissues and/or datasets (**Fig 1**). The key principle

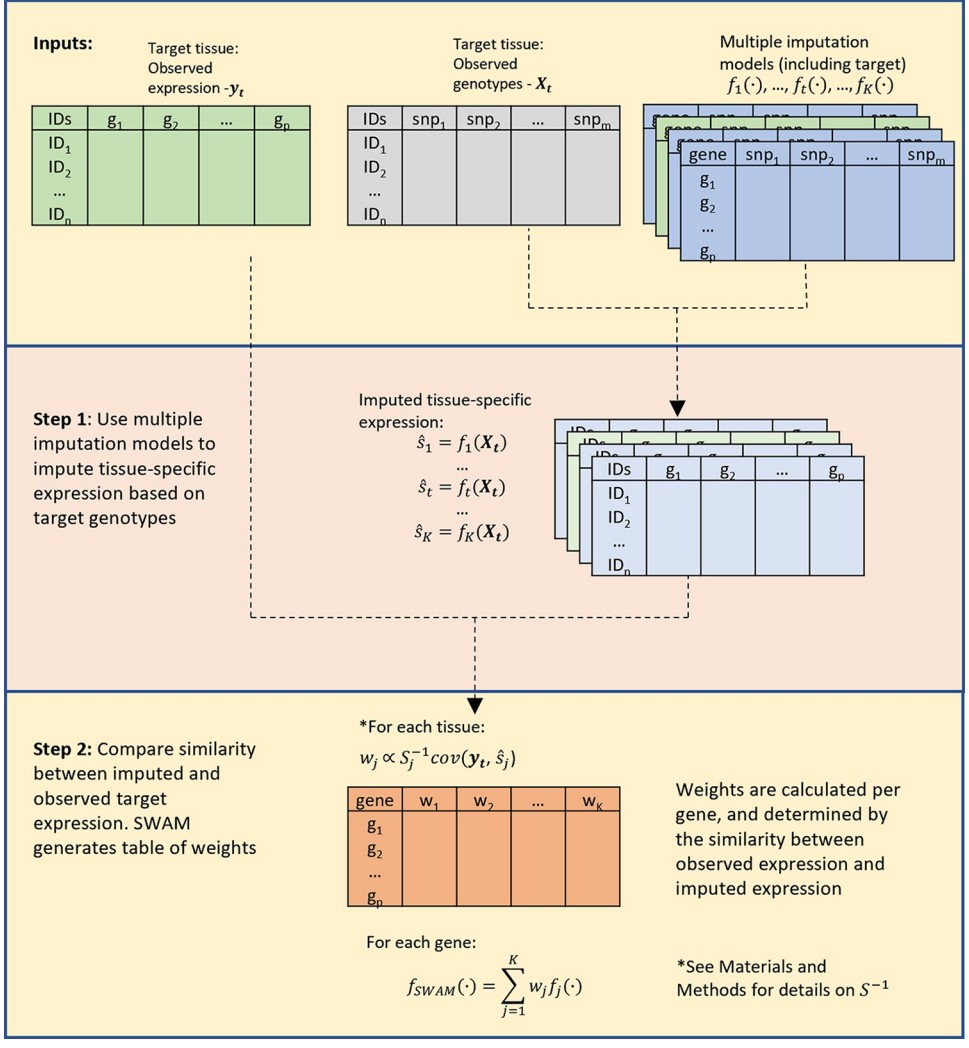

**Fig 1. Overview of SWAM method.** This figure demonstrates the training of the imputation model using the reference data. The inputs required for SWAM are a set of reference genotypes with sample matched measured expression, and the multiple imputation models to be included. The list of multiple imputation models must also include a model derived from the reference data, which can be done via prediXcan. SWAM uses these models to impute tissue-specific expression levels from the reference genotypes. These imputed expression sets are then compared with the measured expression of the reference set. The weights are calculated based on the similarity between the measured and imputed expression and the covariance structure of tissues. For full details, see the Materials and Methods section.

behind SWAM is to improve the accuracy of transcriptomic imputation by determining the optimal linear combination of these multiple imputation models in terms of expected imputation accuracy. To do this, SWAM compares each imputation model to a single reference tissue (tissue of interest) to determine the relative contributions of each imputation model. As such, our method only requires individual-level genotypes and expression for the reference tissue, and then integrates imputation models that were already trained from different tissues and datasets (e.g. GTEx, DGN, and TCGA).

The first step of SWAM is to apply each single-tissue imputation model to the reference genotypes, which results in individual-level, tissue-specific imputed expression. The second step of SWAM compares each imputed expression with the measured expression of the reference tissue to calculate optimal weights by linearly combining multiple imputation models to minimize expected mean squared error (MSE) (see Materials and Methods for the details). The output of second step is an integrated transcriptomic imputation model compatible with the PrediXcan and MetaXcan software tools. Using the SWAM output, we can impute the transcriptome of any samples of interest with genotype information available (via PrediXcan), or to use the model and covariance matrix directly to perform TWAS (via MetaXcan) when GWAS summary statistics are available (S1 Fig).

## Simulation study demonstrates the robustness of SWAM across various scenarios

We performed simulation studies to evaluate SWAM's ability to robustly impute expression by leveraging tissue-specific and cross-tissue components across a wide spectrum of parameter settings. To do this, we independently simulated multi-tissue expression levels along with genotype data for both our training and validation sets (see Materials and Methods). We used the training set solely to derive SWAM models, while the validation sets were used to test our models by comparing the imputed expression with the actual (simulated) expression. We compared SWAM with two heuristic approaches–*naïve average*, which equally weights individual tissue and *best tissue*, which only uses the tissue with the highest expected imputation accuracy–as well as with *single-tissue* imputation and UTMOST.

As expected, we observed *naïve average* to be particularly powerful when the causal variants are shared across all relevant tissues (**Fig 2A**), identifying 93% of genes as significantly imputable at FDR < 0.05. However, when all causal variants were more tissue-specific, the naïve average only identified 19% of genes to be imputable. On the other hand, best-tissue was more powerful (50%) than naïve-average when the all causal variants were tissue-specific, but worse when all causal variants were shared (87%). When only *single-tissue* was used for imputation, the performance stayed similar regardless of the tissue-specificity. Encouragingly, SWAM outperformed all four other methods across all ranges of tissue-specific and cross-tissue heritability settings. We believe this is because SWAM learns tissue-specific weights without preconceptions of tissue relatedness, and thus determines the weights for relevant tissues while ignoring unrelated ones. When we evaluate these methods under the null model, the Type I error rates were well controlled across all five methods (S1 Table).

A similar trend is observed when we vary the number of relevant tissues that shares cross-tissue heritability (**Fig 2B**). In the case where there are no relevant tissues other than the target tissue, naïve average is least powerful while SWAM performs as well as the *single tissue* approach. This suggests that in this scenario, SWAM is correctly giving non-zero weights to only the target tissue, making it similar to the *single-tissue* method. In the other scenario where every tissue is relevant, SWAM provides a similar power to the *naïve average* approach, suggesting that SWAM is robustly assigning weights to each relevant tissue. Similarly, when there

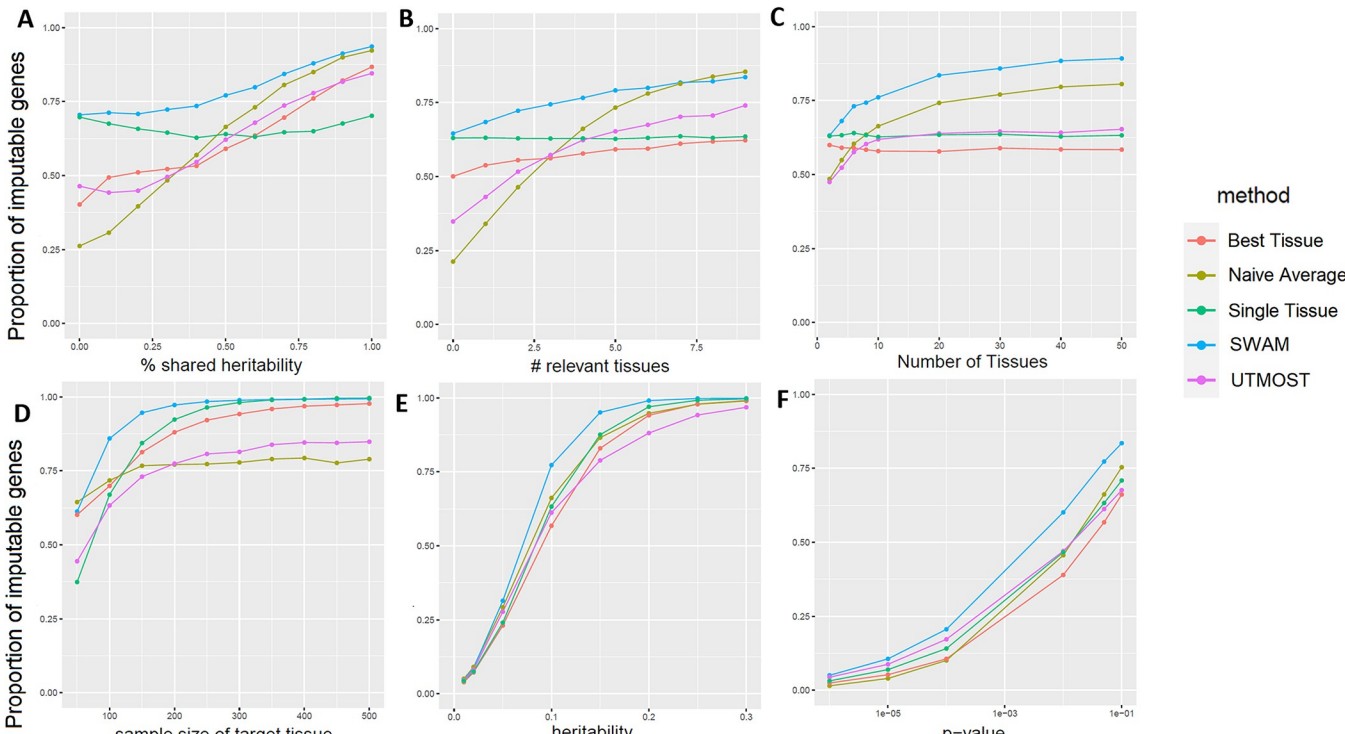

**Fig 2. Simulation study comparing SWAM with naïve average, best tissue and single tissue methods.** We performed each simulation 10,000 times, with the following default settings: 10 total tissues (1 target, 4 relevant, 5 irrelevant), 100 SNPs (2 per tissue), 10% genetic heritability, 50% shared heritability between relevant tissues. In addition, the sample size of the target tissue was 100 individuals, and the remaining tissues had 200 individuals. This was done to emphasize the importance of integrating information from other tissues when the quality of the target tissue model is limited. Five methods–Single Tissue, UTMOST, Best Tissue, Naïve Average, and SWAM were compared. Panel (A) shows the effects of changing the shared heritability for the relevant tissues. We note that each tissue has 10 causal SNPS–for the relevant tissues, 5 of these causal SNPS is shared with the target tissue while the other 5 are independent of all simulated tissues. In panel (B), we varied the number of relevant tissues, from 0 to 10. Panel (C) shows the improvement when the total number of tissues is increased, with the number of irrelevant tissues fixed at 50% of the total. Panel (D) shows the performance of the approaches for different levels of genetic heritability. This simulation demonstrates the range of heritability that we would expect to see the most improvement. Panel (E) shows the effects of target tissue sample size. The x-axis pertains to the sample size of the target tissue only, and all other tissues were fixed at 200 individuals. Finally, panel (F) shows the performance of the methods at different p-value thresholds, using the default simulation settings.

are more tissues available overall (assuming 50% are relevant tissue sharing cross-tissue heritability), the power of SWAM and *naïve average* continues to increase while *single-tissue* and *best-tissue* remain similar (**Fig 2C**). UTMOST demonstrated similar trend to SWAM, but SWAM consistently outperformed UTMOST across all settings.

Our simulation study also evaluated the impact of sample size of the reference tissue. We hypothesized that *single-tissue* would perform poorly when the sample size of the reference tissue was small, which was indeed observed in our results (**Fig 2D**). When the reference tissue has sample sizes of 50, 100, 200, we observed that *single tissue* method identified 39%, 67%, and 93% of imputable genes. Because additional tissues are helpful especially when the reference tissue has smaller sample size, the *best tissue* approach performed better than *single tissue* at lower sample size (60% at n = 50), but worse at higher sample size (88% at n = 200). Similarly, *naïve average* also performed better than *single tissue at* lower sample size (65% at n = 50), but worse at higher sample size (77% at n = 200). However, SWAM consistently outperformed single tissue across all cases (61%, 86%, 97% at n = 50, 100, 200). This implies that borrowing information from a relevant tissue (to the reference) is useful in these situations and SWAM robustly estimates the weights from each tissue accounting for the uncertainty from different sample sizes.

Finally, we investigated the performance of our approaches over different levels of heritability (**Fig 2E**), and across a wide spectrum of p-values (**Fig 2F**), confirming that SWAM outperforms the other methods across these parameter settings. To better reflect more complex and realistic scenarios typically found in eQTL data, we also expanded our simulations to evaluate the impact of a higher number of causal SNPs on predictive performance of SWAM (S2 Fig). Here, we varied the number of causal SNPs from 5 to 125 and examined the performance of SWAM across a wide range of heritability levels. We found that the number of causal variants did not impact the performance of SWAM, with predictive $R^2$ remaining the same when increasing the number of variants. When comparing other methods such as single tissue, naïve average and best tissue, we also found that increasing the number of SNPs from 2 (S7 Fig) to 10 (**Fig 2**) had no meaningful impact.

## SWAM outperforms other transcriptome imputation methods in evaluations with real data by considering the bias-variance tradeoff

We used SWAM to integrate all 49 PrediXcan models trained from each tissue using GTEx v8 release to generate a multi-tissue model. Except for the target tissue LCL (lymphoblastoid cell line tissue or "Cells–EBV-transformed lymphocytes"), individual-level genotypes and expression levels were not used to build the SWAM model. We then externally validated the imputation accuracy of this GTEx-based SWAM model by applying it to impute expression for 344 European LCL samples from the GEUVADIS consortium [29], and compared this to the measured expression levels of the corresponding individuals.

We also evaluated the performance of alternative methods including (1) single-tissue, (2) naïve average, (3) best-tissue, and (4) the *UTMOST* method. For single tissue, we repeated this validation for each of the 49 GTEx v8 single tissue imputation models generated by PrediXcan. Naive average and best tissue, our other heuristic approaches, were also evaluated by using the GTEx LCL as the target tissue. We also evaluated another multi-tissue method *UTMOST* [24]. Because *UTMOST* models were built based on GTEx v6, we used v6 for any evaluations involving *UTMOST*.

Among the single-tissue imputation models, we observed that the imputation from LCL identified 1,620 genes as significantly imputable at FDR < 0.05 (**Fig 3A**). Interestingly, we observed that another tissue, fibroblast cell lines (FCL; the official tissue name in GTEx was "Cells–Cultured fibroblasts"), identified even more genes (2,428 genes) as significantly imputable for GEUVADIS LCL expression levels. One of the outstanding differences between LCL (n = 147) and FCL (n = 483) models were the sample size used for training. We suspect that this is due to (1) the difference in sample size (i.e., FCL imputation has less variance) and (2) the similarity of transcriptomic profiles between LCL and FCL (i.e., FCL model tends not to introduce large bias). However, tissues with larger sample size did not always result in more accurate imputation. When we examined the results from Skeletal muscle model (n = 706), which had the largest sample size in GTEx v8, we identified only 1,762 genes as significantly imputable. This is likely because the large differences of transcriptomic profiles between LCL and Skeletal muscle (i.e., Skeletal muscle model tends to introduce large bias). These examples demonstrate that both sample size and tissue relevancy are important for maximizing the imputation accuracy. In statistical terms, our primary interest was to reduce the mean-squared error (MSE), which is the sum of $Bias^2$ and Variance. We suspect that FCL model performed better than LCL models due to much smaller variance (because of larger sample size), and better than Skeletal muscle models due to much smaller bias (S3 Fig). We hypothesized that by combining imputations from multiple models, we can minimize MSE by substantially reducing variance without introducing excessive bias, which was our main motivation for

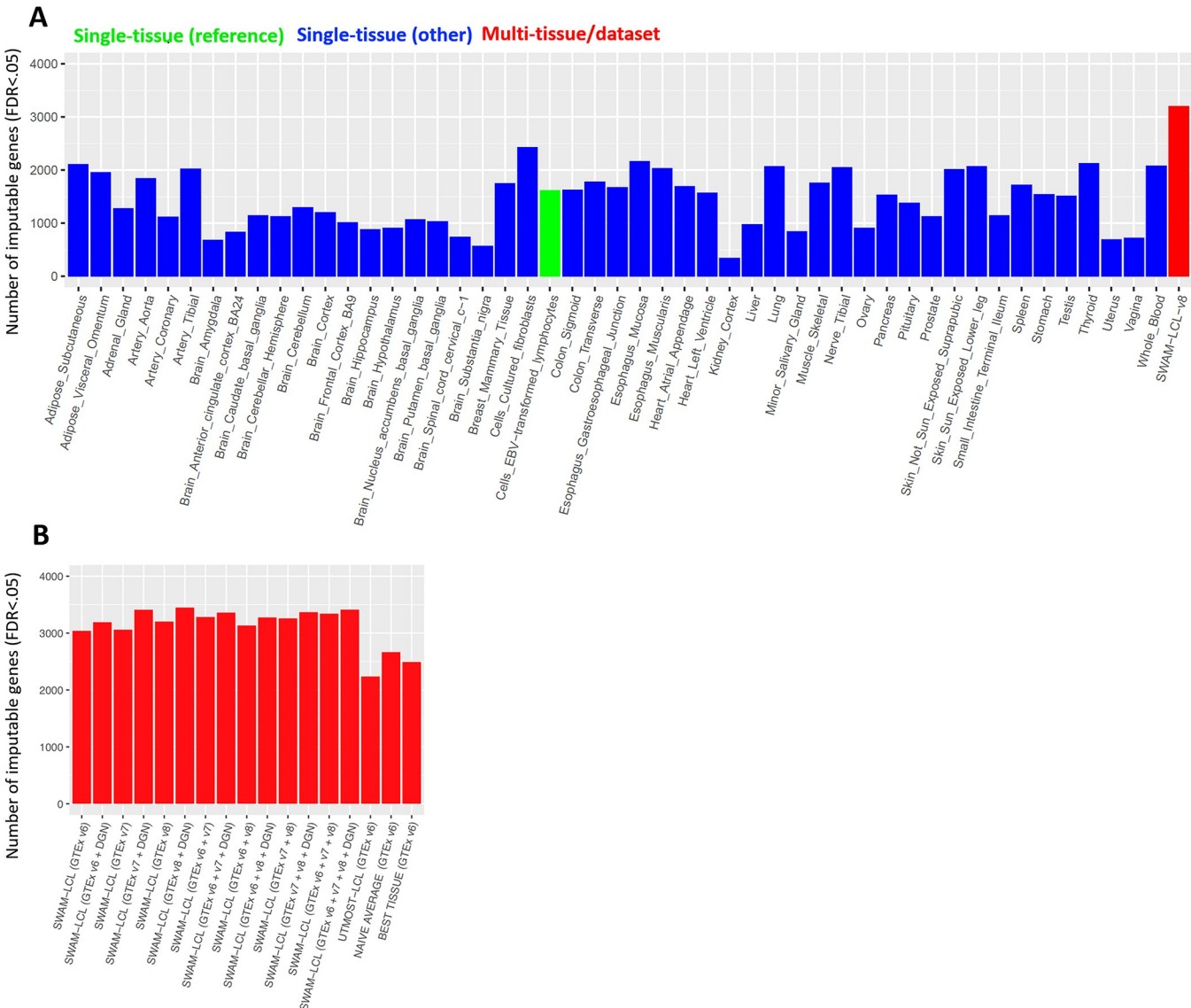

**Fig 3. Empirical validation of SWAM using lymphoblastoid-cell line data from GEUVADIS consortium.** We used our LCL-targeted SWAM model to impute expression levels based on the genotypes of 344 European samples. We then calculated the concordance between imputed expression and measured LCL expression. We repeated this for all of the other methods mentioned here. (A) shows the performance of SWAM against the single-tissue models from 49 tissue-specific predictDB models derived from GTEx version 8. In (B), we derived various SWAM models using every combination of the following: 1) all GTEx v6 tissues, 2) all GTEx v7 tissues, 3) all GTEx v8 tissues, and 4) Depression Gene Network (DGN) single tissue whole blood model from predictDB. Here, we also included the UTMOST LCL model, naïve average and best tissue models, all derived from GTEx v6.

developing SWAM. When comparing SWAM to the single-tissue models, our LCL-targeted SWAM approach identified 3,203 genes that were significantly imputable at FDR < 0.05, a 97.7% over the predictDB LCL model, and a 31.2% increase over the best performing tissue (FCL).

When evaluating the multi-tissue methods, our two heuristic approaches (both using GTEx v6 tissues), *best-tissue* and *naïve average* identified 2,493 and 2,666 significantly imputable genes, respectively, which was >47% and >57% higher than any of the 44 GTEx v6 single tissue models. *UTMOST* (using the LCL model) also substantially increased the number of

imputable genes (2,238 genes, >32% increase over any single tissue from GTEx v6), but sur-prisingly, it had fewer than the imputable genes compared to the two heuristic approaches. Finally, when we applied *SWAM* specifying GTEx-v6 LCL as the reference tissue, the number of imputable genes further increased to 3,040, which is >79% larger than any other single tis-sue models (S2 and S5 Tables). Interestingly, *SWAM* improved the imputation accuracy over *UTMOST* even though it requires individual-level data only for one tissue (i.e., LCL) in GTEx while *UTMOST* requires simultaneous access to individual-level data across all tissues (**Fig 3B**). These results demonstrate that SWAM offers an accurate and flexible meta-imputation framework by optimally combining multiple imputation models across tissues.

## SWAM enables meta-imputation of expression levels across multiple heterogeneous datasets

One of the important advantages of SWAM compared to other multi-tissue imputation meth-ods is the ability to integrate imputation models across heterogeneous datasets where samples may not necessarily overlap. To evaluate the benefit of SWAM's ability for multi-dataset "meta-imputation", we integrated imputation models trained from GTEx v7 and v8, as well as 922 whole blood transcriptomes from Depression Gene Network (DGN). The rationale to include GTEx v7 and v8 models (S3–S4 Tables) is that the datasets are slightly different from v6 (for example, v7 has more samples in all tissues except for LCL, FCL, and whole blood) and integrating multiple training models from slightly different versions of datasets may improve the accuracy. The reason to include DGN whole blood is that the sample size is much larger than any individual tissue GTEx, so it may help further reduce the variance and MSE of the imputation model.

   When applying SWAM to GTEx v6, v7, or v8 datasets individually, the number of signifi-cantly imputed genes at FDR < .05 were 3,040, 3,060, and 3,203, respectively (**Fig 3B**). How-ever, when all datasets were combined, the number of imputable genes increased to 3,342. These results suggest that imputation across multiple datasets can help even when the datasets are highly overlapping. When we additionally integrated SWAM with the DGN whole blood model, which detected 2,390 imputable genes by itself, the number of imputable genes by the integrated SWAM model further increased to 3,413. Note that we needed individual-level data only for the reference tissue/data (GTEx v6 LCL in our experiment), so an arbitrary combina-tion of imputation models, which consist of only summary-level data, can be seamlessly added to the meta-imputation framework of SWAM.

   Overall, using all 49 GTEx v8 tissues in combination with the DGN whole blood model pro-vided the highest number of imputable genes, with a 112.9% improvement over the corre-sponding GTEx v8 PrediXcan-LCL model (single tissue), and a 13.5% improvement over the GTEx v6 version of SWAM-LCL (multi-tissue) (**Fig 3B**). Regardless of the version of GTEx used, including the DGN whole blood model gives a substantial improvement in number of imputable genes compared to not including it in the model. Another interesting observation is that while PrediXcan-LCL (v6) appears to perform better than PrediXcan-LCL (v7), SWAM-LCL derived from v7 performs better than v6 SWAM-LCL. This may suggest that while GTEx v7 PrediXcan-LCL may not have had a significant improvement in eQTL detec-tion compared to its predecessor, other tissues may have improved in more substantial ways. This is because the sample size for LCL in v7 decreased by 18 samples, whereas other non-blood tissues had substantial sample size gains of up to 89 individuals. Here, SWAM leverages the increase in quality from other tissues, which allows for better overall imputation regardless of the quality of the target tissue itself.

## SWAM robustly captures both tissue-specific and cross-tissue regulatory components

The key component behind the robust performance of SWAM is that it learns how to distribute weights across multiple imputation models for each gene individually. If a gene shares eQTLs across many tissues, the SWAM's weights will be distributed evenly across tissues and the model will behave similarly to the naïve average heuristic. For example, *ERAP2* is a well-known gene with shared eQTLs profiles across most tissues. In GTEx v8, *ERAP2* can be reliably imputed with any of the 49 single-tissue imputation models from PrediXcan with $r^2 >$ 0.77 or higher. As a result, the weights from SWAM are almost evenly distributed across the tissues, ranging from 0.006 (0.01 excluding LCL) to 0.031 (S4 Fig), and the accuracy of SWAM ($r^2 = 0.812$) is very similar to the accuracy of naïve average ($r^2 = 0.811$).

On the other hand, when the imputation model from the reference tissue is not particularly good due to smaller sample size or other technical issues, SWAM can substantially improve accuracy by leveraging eQTL sharing from other tissues. For example, the single-tissue imputation accuracy of *GSTM4* is relatively low in LCL tissue ($r^2 = 0.125$) compared to the accuracy of the 38 other tissues in which a PrediXcan imputation model is available (average $r^2 = 0.307$). Using SWAM, leverages 39 tissues by assigning low weight to the LCL tissue, and higher weights to more relevant tissues, which thereby increases the predictive R-squared to $r^2 = 0.492$ (S4 Fig).

Finally, for genes that are highly tissue-specific, the SWAM's weights will be distributed similarly to the best tissue heuristic. For example, *PTBP3* is expressed in most tissues, but has significant eQTL signals in only 15 tissues. SWAM assigns weights to 9 of these tissues, and substantially improves the predictive accuracy from $r^2 = 0.111$ to $r^2 = 0.447$ (S4 Fig).

## Comparison of imputation models in the context of TWAS

We conducted TWAS analysis using SWAM, UTMOST, and PrediXcan models via MetaXcan [30]. In addition, we also used S-MultiXcan [25] to simultaneously test all of the PrediXcan models using their PCA regression approach. To control for genomic inflation, we calculated and reported the genomic control inflation factor ($\lambda_{GC}$) for each TWAS dataset (S9 Table). We found that the inflation factor was very similar across methods (S8 Fig), with medians of $\lambda_{GC} =$ 1.11 for prediXcan, $\lambda_{GC} = 1.24$ for UTMOST and $\lambda_{GC} = 1.15$ for SWAM. We used a Bonferroni correction to establish p-value threshold for each analysis separately, based on the number of genes imputed. Overall, we found that among the methods that directly estimate expression levels (SWAM, UTMOST, PrediXcan), SWAM outperformed the other methods in terms of number of associations detected (see S6–S8 Tables). For example, PrediXcan models on average detected 20.5, 20.4 and 3.0 transcriptome-trait associations for HDL, LDL and T2D respectively. For SWAM, we observed an average of 66.7, 67.7 and 4.6 associations per tissue, whereas UTMOST yielded an average of 53.7, 53.5 and 3.9 associations per tissue, for the three traits respectively.

We also examined the number of replicated signals between the two multi-tissue approaches (SWAM, UTMOST) and the single-tissue method (PrediXcan). We found that on average, 17.4%, 17.4% and 33.3% of SWAM's signals were replicated in their single-tissue counterpart for HDL, LDL and T2D respectively. For UTMOST, these replication rates were 13.9%, 12.1% and 23.7% for HDL, LDL and T2D. The higher overlap in signals for SWAM models may be a result of higher tissue-specificity compared to the UTMOST models.

We plotted transcriptome-wide signals for the LDL trait using the GTEx v6 liver model for PrediXcan, UTMOST and SWAM (**Fig 4**). One interesting signal gained from the SWAM analysis is the *APOC1* gene, which is primarily expressed in the liver and has been implicated in playing a role in HDL and LDL/VLDL (very low-density lipid) metabolism [31]. For this

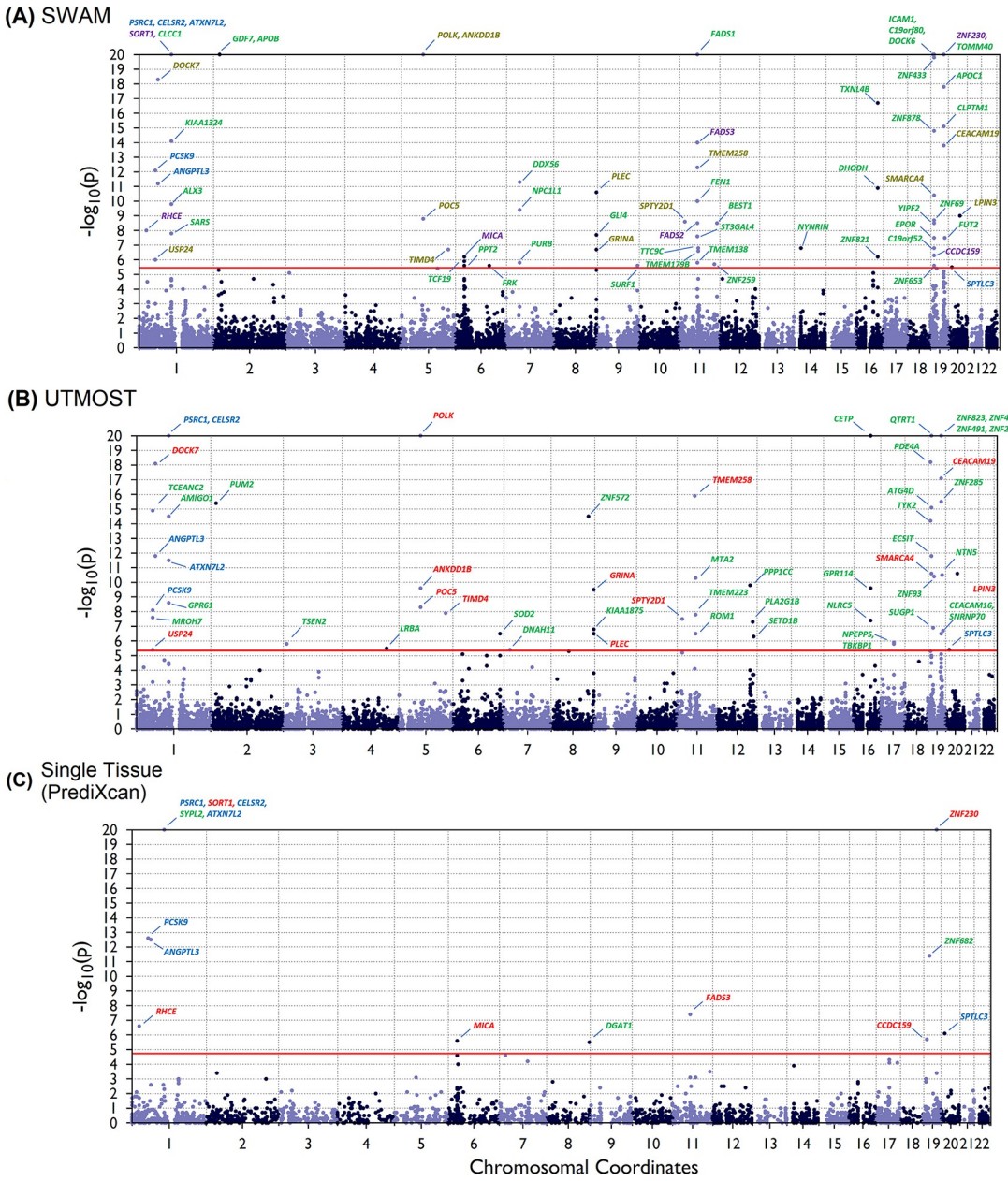

**Fig 4. TWAS on LDL trait targeting liver using SWAM, UTMOST and PrediXcan models.** TWAS was performed using metaXcan on the LDL trait from the Global Lipids Genetics Consortium (GLGC) GWA analysis. For a consistent comparison, the SWAM and UTMOST models were derived from GTEx version 6 tissues, and the prediXcan model used was GTEx v6 liver. The number of associations were: 66, 56 and 15 for SWAM, UTMOST and prediXcan respectively. P-values were capped at $10^{-20}$ in these plots.

trait-tissue combination, every gene replicated between UTMOST and PrediXcan was also replicated between SWAM and PrediXcan. In addition, the *SORT1* gene, which has been found to be associated with LDL-C levels, was only detected in SWAM and PrediXcan [32].

One potential shortcoming for both multi-tissue approaches (SWAM and UTMOST) appear to be that the number of unique signals (across all tissues) is fewer than those generated

by PrediXcan's single tissue models. For example, SWAM produces 210 unique associations for the HDL trait, while we see 187 unique associations from UTMOST and 248 unique associations from PrediXcan. Similarly, MultiXcan detects 284 significant associations when scanning across all tissues (based off the PrediXcan models). It appears that while the multi-tissue methods can leverage information from other tissues to impute expression accurately, marginal association signals in TWAS are potentially lost using these approaches. However, we found that a high number of these unique signals from the PrediXcan TWAS appeared only in one or two tissues (92.5% for HDL, 98.2% for LDL and 100% for T2D).

With all these various considerations, SWAM appears to improve TWAS power for a given tissue, although ultimately may yield fewer signals compared to comprehensive tissue scans using PrediXcan or MultiXcan. While SWAM outperforms other methods in terms of imputation accuracy, there may not be a clear-cut winner in terms of performance in TWAS. The best approach to use will likely depend on the needs of the researcher, and each approach may provide different yet complementary insights into understanding the biological mechanisms from these association studies.

## Discussion

The transcriptome serves as an intermediate phenotype linking genetic variants to complex traits. Association studies between traits and gene expression, when used in conjunction with GWAS, provide additional insight into the biological mechanisms of complex traits. Imputation of gene expression in the context of transcriptome wide association studies is a promising approach to understanding the connection between our genes and many traits. Yet, there are still many challenges that arise when performing association studies with imputed expression. Current tissue-specific imputation models are trained using data obtained from their respective tissues, which can vary greatly in data quality and sample size. As such, there is a great deal of variability among tissues in the imputation accuracy of tissue-specific gene expression levels. For example, PrediXcan was able to significantly impute only 1,919 vagina-specific genes, while it discovered 7,764 genes specific to the tibial nerve tissue. Furthermore, the imputation accuracy of significant genes within a tissue are also highly variable, with some genes such as ERAP2 having very high (>80% of variation explained by eQTLs) imputability and other genes (~1% of variation explained by eQTLs) with low imputability.

In this paper we developed SWAM, a method that determines the level of eQTL sharing between tissues and uses the shared information from other tissues to improve the imputation accuracy for the target tissue. By simultaneously examining the relatedness of multiple tissues, SWAM in essence increases the effective sample size of imputation models. Using GEUVADIS LCL data, we compared SWAM to single-tissue approaches. We found that our multi-tissue approach, in addition to increasing the number of significantly imputable genes for each tissue, also improved the overall imputation accuracy for genes that were already significantly imputable using PrediXcan. We improved the power of TWAS by running a SWAM-adapted version of MetaXcan for various traits, finding an increased number of significant transcriptome-trait associations, even when correcting for the larger number of genes imputed.

Although SWAM provides a substantial improvement for the number of significantly imputable genes for many tissues and generally increases power for TWAS, there are some shortcomings and caveats to consider with the approach. It is important to note that unlike PrediXcan, SWAM does not actually perform model training or eQTL discovery. Instead, it evaluates the efficacy of various single-tissue imputation models (in this case, the GTEx tissues) and assigns weights to the models based on their relatedness to the target tissue. Therefore, for SWAM to work, there must already be a database of imputation models that it can use

to derive the multi-tissue weighting. Because we are utilizing existing imputation models, we acknowledge that there will be cases where the SWAM imputation accuracy could be similar or worse to the single-tissue imputation, especially if the gene has shared eQTLs across many tissues or if the single-tissue imputation model was already performing well. The improvement observed in our validations and TWAS are an overall trend, and as with any analysis, interpretation of any specific results should be approached with caution. Furthermore, the improvement for any given gene has an upper limit which is dependent on the pool of single tissue models available. There may be tissues that have very few relevant other tissues to draw information from. For any given gene within the target tissue, SWAM automatically assigns weights of non-relevant tissues to zero based on a threshold. However, for the purposes of our study, the threshold was tuned to be more lenient, allowing for more tissues to be included in the imputation of each gene's expression levels. A more lenient threshold will yield more genes, but a lower sensitivity to the target tissue. A stricter threshold will provide imputed expressions that are more specific to the target tissue but will provide imputation for fewer genes and may reduce imputation accuracy in some genes. Similarly, the choice of regularization parameter λ may need a fine-tuning depending on the correlation structure between tissues. In such cases, the SWAM software tool offers automatic selection of the parameter λ using cross-validation. Further in-depth analysis of the behavior of SWAM imputation model, which is essentially an ensemble learning method across multiple imputation models, could help determine these thresholds more efficiently without having to rely on computationally intensive procedure to select these parameters empirically.

Next, our empirical validation of imputation accuracy was tested on European individuals (344 samples from GEUVADIS) and thus SWAM's performance with other populations has not yet been determined. A future direction of research could be to examine whether a single model derived from mixed populations would represent each of the populations accurately, or if a different model should be trained on each population separately. Currently, evidence suggests that training from the correct ancestry group is the ideal approach for population-specific imputation [33], which emphasizes the importance of reference panel resources derived from a wide array of ancestries. Alternative approaches could be to leverage trans-ancestry correlation, which has been shown to increase predictive $R^2$ in the context of polygenic risk scores [34].

Finally, while SWAM improved the number of association signals for any given tissue in TWAS compared to UTMOST and single tissue PrediXcan, aggregation of signals (MultiXcan/combining PrediXcan signals) suggest that other approaches may yield more unique signals. It is unclear which approach is preferable in this scenario, and the answer may depend on unraveling the causality of association signals. Recently, there have been a number of publications which have addressed this issue, such as PTWAS which uses instrumental variables (IVs) to investigate the causal relationship between expression levels and complex traits [26], or phenomeXcan, which integrates GWAS and gene expression and regulation data to identify likely causal pathways [35]. Future directions could include using IVs or functional annotation to interpret TWAS signals.

While SWAM did not demonstrate inflation of false positives in our simulation, the genomic control inflation factor lambda for LDL, HDL, and T2D were higher than 1.0 across all TWAS methods we evaluated. We believe that this is not because of false positives but due to combined effect of large sample size and linkage disequilibrium (LD) between causal variants and imputed expression levels. In principle, the LDSC method [36] can help disentangle the effect from false positives and pervasive LD if it supported results from TWAS association using the intercept term. We note that this topic of quantifying the true inflation of TWAS statistics by accounting for LD score as a future research topic of interest.

To conclude, we propose a novel method for gene expression imputation, which extends already established single-tissue imputation models into a multi-tissue setting. By combining

information from multiple models, we were able to increase overall tissue-specific imputation accuracy for many genes and increase power for transcriptome-wide association studies.

## Materials and methods

### SWAM notation and framework

Our framework for *SWAM* is designed to find the optimal linear combination of imputed expression levels from multiple tissues and datasets. For simplicity, we will denote each (tissue, dataset) combination as a source. We assume there are $K$ imputation models from individual sources, with each model indexed as $j \in (1,..,K)$. We also denote $t \in \{1,...,K\}$ to represent the index of the reference source (which is the target tissue). The inputs for *SWAM* are: (1) $f_j(\cdot)$– the single-source imputation models and (2) $Y_t$ and $X_t$–the individual-level gene expression measurements and genotypes for the reference source. For each gene $g$, let $\hat{s}_j^g = f_j(X|g)$ be imputed expression from a single source. Then we can represent any linearly combined multi-tissue imputed expression $\hat{m}_t^g$ as

$$\hat{m}_t^g = \sum_{j=1}^K w_j^g \hat{s}_j^g$$

where $w_j^g$ is the weight contributed by $j$-th source. *SWAM* learns $w_j^g$ by leveraging individual-level data from the reference source as we describe later.

### Multi-tissue methods using naïve average or best-tissue

There are two heuristic approaches to impute expressions from multiple sources—*naïve average* and *best tissue*. *Naïve average* defines weights uniformly as $w_1^g = \ldots = w_K^g = {}^1/_K$. For *best tissue*, the weights are defined as a dichotomous variable:

$$w_j^g = \begin{cases} 1 & \text{if } j = \text{argmax}_i(cor(\hat{s}_i^g, y_t^g)) \\ 0 & \text{otherwise} \end{cases}$$

where $y_t^g$ represents the individual-level expression measurements of the reference source.

### Smartly Weighted Average across Multiple Datasets (SWAM)

Here we describe how SWAM calculates optimal $w_j^g$, whose derivation is shown in the S1 Text. It is important to note that *SWAM* works ideally when the tissue type intended to be imputed matches to the tissue types of the reference source. We define $y_t^g$ as the $n \times 1$ vector of individual-level expression measurements for the reference source, and as before, $X_t$ to be the corresponding $n \times m$ matrix of individual-level genotypes. The first step is to impute expression using each of the $K$ models using the reference genotypes. Thus, we obtain $K$ sets of imputed expressions, $\hat{s}_j^g = f_j(X_t|g)$, with each being a single-source imputation for the samples in the reference data. The weights for *SWAM* are given by

$$\boldsymbol{w^g} = (w_1^g, w_2^g, \ldots, w_K^g)^T = \left[ \begin{bmatrix} cor(\hat{s}_1^g, \hat{s}_1^g) & \cdots & cor(\hat{s}_1^g, \hat{s}_K^g) \\ \vdots & \ddots & \vdots \\ cor(\hat{s}_K^g, \hat{s}_1^g) & \cdots & cor(\hat{s}_K^g, \hat{s}_K^g) \end{bmatrix} + \lambda I \right]^{-1} \begin{bmatrix} cor(\hat{s}_1^g, y_t^g) \\ \vdots \\ cor(\hat{s}_K^g, y_t^g) \end{bmatrix}$$

Here, the correlation matrix, which is represented as $S$ in Fig 1, account for the similarity between the imputation models, and the vector containing the entries $cor(\hat{s}_j^g, y_t^g)$ account for

the empirical similarity of imputed expressions from each model to the measured expressions in the reference source. When $j = t$, because $cor(\hat{s}_t^g, y_t^g)$ will be prone to overfitting, we replace this value to a 5-fold cross-validated correlation instead, which is available from PrediXcan output. Finally, $\lambda I$ acts to regularize the weights, providing numerical stability for the inversion of the covariance matrix. The calibration of $\lambda$ is further discussed in the S1 Text.

## Simulations

Our simulation study sought to examine SWAM's ability to detect the correct shared components between related tissues across a wide spectrum of parameter settings. We compared *SWAM* with *naïve average*, *best tissue* and *single tissue* approaches. For each simulation, we independently generate individual-level genotypes and expression multiple tissues. For the reference set, we simulated $X_r$, an $n_r \times m$ genotype where $n_r$ is the number of individuals and $m$ the number of SNPs. In our simple simulation, we assume that each SNP is independent, with non-reference allele frequency (AF) distributed with *Beta(1,3)*. The genotypes were simulated using a binomial distribution based off the AF. To simulate multi-tissue expressions, for each tissue $j \in (1,..,K)$ we specific effect sizes $\boldsymbol{\beta_j}$, to simulate expressions $\boldsymbol{y_j} = X_t\boldsymbol{\beta_j} + \varepsilon_j$. For reference tissue (i.e. $j = t$), we assume ten causal SNPs with nonzero elements in $\boldsymbol{\beta_j}$, where five SNPs is expected to explain tissue-specific heritability ($h_t^2$) for the reference tissue and the other five SNPs explain the cross-tissue heritability ($h_c^2$), summing up to total heritability ($h^2 = h_t^2 + h_c^2$). Other tissues (i.e. $j \neq t$) were divided into "related tissues" and "independent tissues". For related issues, $\boldsymbol{\beta_j}$ had only one non-zero value corresponding to cross-tissue heritability ($h_c^2$). For independent tissues, all $\boldsymbol{\beta_j}$ had zero values. Finally, we generated another set of validation genotypes matrix $X_v$ with size $n_v \times m$, and the validation expressions ($\boldsymbol{y_v} = X_v\boldsymbol{\beta_t} + \varepsilon_v$) of reference tissue using the same settings to use for evaluation.

We then trained tissue-specific imputation models $f_j(.), j \in (1, \ldots, K)$ by applying an elastic-net model (using *glmnet* R package [37]) for each pair of $X_t$ and $\boldsymbol{y_j}$. The tuning parameters for elastic net were determined via a five-fold cross-validation technique. Using $\boldsymbol{y_t}$, $X_t$ and $f_i(.)$, we obtained *naïve average*, *best tissue* and SWAM models as detailed in the previous section. Evaluation with UTMOST was implemented with CTIMP package (https://github.com/yiminghu/CTIMP) so that it performs the same as UTMOST but handles our simulation data, as suggested by the authors (https://github.com/Joker-Jerome/UTMOST/issues/12). The code for simulation and evaluation is publicly available at https://github.com/aeyliu/SWAM-manuscript. To calculate the proportion of imputable genes, we performed linear regression between $\boldsymbol{y_v}$ and the imputed expression from genotypes $X_v$ using the different methods to obtain a p-value.

Each simulation was repeated for 10,000 times in each setting. We varied parameters to evaluate their impact on the performance of each method. We varied $h^2 \in \{0, 0.1, \cdots, 1\}$ (default 0.1), $h_c^2/h^2 \in \{0, 0.1, \cdots, 1\}$ (default 0.5), $K \in \{2, 4, 6, 8, 10, 20, 30, 40, 50\}$ (default 10), fraction of independent tissues ranging $\{0, 0.1, \cdots, 0.8\}$ (default 0.5), $n_r \in \{50, 100, \cdots, 500\}$ (default 200), and the p-value threshold ranging $\{10^{-6}, 10^{-5}, \cdots, 0.01, 0.05, 0.1\}$ (default 0.05). Throughout all simulations, $m = 35$, $n_v = 200$ were used.

We then expanded our simulation to include a larger number of causal variants, ranging from 5 to 125 SNPs across a spectrum of heritability levels. For these simulations, we kept most the same parameter settings as our other simulations, except for generating 5,000 individuals as opposed to the default 200.

## Input datasets: Genotypes, expressions, and imputation models

In our experiments with real datasets, we leveraged multiple published datasets where genotypes, expressions, and imputation models are available to evaluate the performance of SWAM

and other methods in various settings. Specifically, we used the GEUVADIS LCL [29] genotypes and expressions as a validation dataset. We used GTEx data [14] [24] and PredictDB [2] to build multi-tissue imputation models. To demonstrate the ability to SWAM to incorporate multiple datasets, we used DGN [11] dataset as well as multiple versions of GTEx datasets.

## Multi-tissue transcriptomic profiles and imputation models from the GTEx project

To build multi-tissue imputation models using *SWAM*, *UTMOST*, *naïve average*, *and best tissue methods*, we used single-tissue imputation models, individual-level genotypes, and expressions obtained from the GTEx consortium. Single-tissue imputation models were downloaded from the PredictDB (http://predictdb.org/) repository for GTEx versions 6, 7 and 8 (44, 48 and 49 tissues respectively) [3] [14] [24], which were trained using PrediXcan's elastic net methods. Individual-level genotypes and expression levels were only used for the reference tissue (e.g. EBV-transformed lymphocytes) which is deemed to be the closest to the validation data (e.g. GEUVADIS LCL), using GTEx version 6.

When evaluating multi-tissue imputation models within a single dataset, we used GTEx version 6. When evaluating imputation models across multiple tissues and multiple datasets, we used various combinations of GTEx versions to evaluate the benefit of multiple imputation models trained from overlapping datasets. When training across different datasets, genes were matched by ensemble ID, ignoring version numbers. In addition to training SWAM, we also used the *single tissue* PredictDB imputation models as a basis for comparison with our method.

## Validation dataset from the GEUVADIS study

We used individual-level genotypes and expression levels from lymphoblastoid cell lines (LCL) from the GEUVADIS consortium only to evaluate various methods after imputing expression levels with models built from other datasets. Each imputation model was evaluated by applying the model to GEUVADIS genotypes to impute individual expression levels, and by calculating the correlation between the imputed and measured expressions. We focused on 344 European individuals where genotypes and normalized expressions (from RNA-seq) are available, with comparable linkage disequilibrium (LD) structure to GTEx and DGN datasets.

## Imputation models from Depression Genes Network

We also downloaded the imputation model trained using the 922 whole blood transcriptomes from the Depression Genes Network (DGN) via PredictDB. DGN was evaluated as a single-tissue imputation model. It was also used in the evaluation of multi-dataset imputation models when DGN is combined with various versions of GTEx imputation models.

## Imputation models from UTMOST

We compared our methods to *UTMOST*, another multi-tissue approach for expression imputation [24]. The *UTMOST* imputation models were jointly trained across 44 tissues from GTEx version 6 and were downloaded from their published online repository (https://github.com/Joker-Jerome/UTMOST). We applied the imputation model targeted for EBV-transformed lymphocytes when evaluating the imputation accuracy with the GEUVADIS LCL expression.

### Evaluating imputation accuracy with GEUVADIS measured expression

We evaluated the accuracy of various imputation models by comparing imputed expressions from individual-level genotypes with the measured expression from GEUVADIS LCLs. Individual-level expression were imputed across 344 European GEUVADIS samples using various single-tissue, multi-tissue/multi-dataset methods to calculate the correlation with the normalized measured expression from GEUVADIS LCL. The correlation between imputed and measured expressions were calculated using spearman correlation and a one-sided p-value was evaluated by converting the correlation coefficients into t-statistics. Genes were considered "significantly imputable" if the Benjamini-Hochberg false discovery rate (FDR) was less than 0.05. This procedure was applied across all genes within each method, with the counts being tabulated.

### Comparing single-tissue and multi-tissue imputation models within a single dataset

With these results, we first focused on comparing the imputation accuracy of SWAM with other methods using GTEx v6, v7 and v8. We compared *SWAM-LCL* (SWAM using GTEx EBV-transformed lymphocytes as reference), every *single tissue* imputation model from PredictDB, *UTMOST-LCL* (*UTMOST* using GTEx EBV-transformed lymphocytes as reference), *naïve average*, and *best tissue* methods. We focused on evaluation using GTEx v6 models where *UTMOST* models were available. We also focused on genes included in the Consensus Coding Sequence Project (CCDS) [38] to minimize the discrepancy between imputation models.

To keep a fair comparison with *UTMOST* and the *single tissue* methods, we restricted the set of genes to those that have at least one eQTL in any *single tissue* models from PredictDB and also in any *UTMOST* models across all reference tissues.

### Evaluating multi-tissue imputation models across multiple datasets

Our second comparison was conducted to examine the effect of integrating multiple imputation models trained from heterogeneous datasets into SWAM. Here, we used various combinations of GTEx and DGN resources to derive multi-tissue/multi-dataset models, such as combining GTEx v6 with DGN data, or combining GTEx v6, v7 and v8 altogether. For this analysis, the gene list was restricted to genes that were included in all three of the v6, v7 and v8 datasets in terms of Ensemble IDs.

### Evaluation of SWAM in transcriptome-wide association studies (TWAS)

To evaluate our method in the context of TWAS, we used MetaXcan [30], which infers TWAS results from GWAS summary statistics. We focused on the HDL and LDL traits from Global Lipids Genetics Consortium (GLGC) [39] and Type-2 Diabetes (T2D) from the DIAGRAM consortium [40]. For this analysis, we generated SWAM imputation models targeting each of the 44 tissues from GTEx version 6. We used MetaXcan to infer the TWAS results for each of these tissues and applied a Bonferroni correction with false-positive rate of 0.05 based on the number of genes tested. We repeated this with all 44 UTMOST models as well as all 44 PrediXcan *single tissue* models.

We also compared our method with S-MultiXcan [25], a recently published extension of MetaXcan which uses a principal components regression to conduct trait-expression association with multiple tissues.

## Supporting information

**S1 Text. Supplementary Text.**
(DOCX)

**S1 Fig. Using SWAM to impute expression and conduct TWAS.** The first panel shows how SWAM can be used to impute expression levels via prediXcan, while the second panel shows the required inputs to conduct TWAS via metaXcan.
(PDF)

**S2 Fig. Impact of number of causal variants in simulation study.** We expanded our simulation study to examine the effects of a larger number of causal variants, ranging from 5 to 125 variants, across a wide range of heritability levels. We found that increasing the number of causal variants had very little effect on the predictive performance of SWAM. We believe that this is because the expected imputation accuracy largely depends on the total heritability explained by the causal SNPs.
(PDF)

**S3 Fig. Bias-variance tradeoff for other tissues.** The principal behind SWAM is it considers the bias-variance tradeoff for each tissue, and assigns higher weights to tissues that reduce MSE. In this example, tissues such as Skeletal Muscle have a high sample size (and therefore lower variance) but may be biased as they are not the relevant tissue to the tissue of interest (in this case LCL). Other tissues such as Fibroblasts may have a lower sample size but compensate by having low bias (high relevance to tissue of interest) and will contribute more weight.
(PDF)

**S4 Fig. The distribution of weights for SWAM for three selected genes.** (A) shows the ERAP2 gene, which had a single tissue $r^2 = 0.854$, while the SWAM model had $r^2 = 0.812$. (B) depicts the scenario where SWAM is able to leverage information from other tissues to make up for the relatively lower quality of the target tissue. here the single tissue model gave $r^2 = 0.125$ while SWAM increased the accuracy to $r^2 = 0.492$. (C) shows an example where the eQTLs are highly tissue specific. Here, SWAM improved the single tissue accuracy from $r^2 = 0.077$ to $r^2 = 0.323$.
(PDF)

**S5 Fig. Calibration of tuning parameter for SWAM using empirical data.** We calibrated our tuning parameter using GEUVADIS data as our external validation. As a result, we set the default value of this tuning parameter to 3 in our software, which we believe performs best. However, as the parameter may depend on the scaling and normalization of the data, there is also an option to calculate the tuning parameter via cross-validation.
(PDF)

**S6 Fig. Distribution of SWAM weights in imputation models for all 44 GTEx v6 tissues.** Here, we used SWAM to derive multi-tissue imputation models for all 44 GTEx v6 tissues. Each cell in this heatmap depicts the number of times each tissue contributed the highest weight to the target tissue. Here, the rows correspond to the target tissue and the columns correspond to the weight contribution of each tissue. For the sake of clarity, the diagonal values were not included as they were consistently much higher than the remaining elements of the matrix.
(PDF)

**S7 Fig. Simulation study with only 2 causal variants.** We performed a comprehensive simulation study identical to the one displayed in Fig 2, but with only 2 causal variants instead of

10. We find that the adjusting the number of causal variants does not affect the results in any substantial way, which we believe is due to the fact that heritability was fixed regardless of the number of variants used.
(PDF)

**S8 Fig. Q-Q plot of p-values for Liver tissue in LDL and T2D TWAS.** Here we display the Q-Q plot of p-values before and after controlling for genomic inflation in our Liver-LDL and liver-T2D TWAS. The LDL plot suggests an enrichment in signals due to high power in the original GWAS analysis.
(PDF)

**S1 Table. Evaluation of Type I errors across different methods using simulated expression and genotype data.** We tested type 1 error rates by training different methods on simulated data and comparing predicted expression concordance with simulated null test data (gene expression generated with zero genetic effect). Here, we show the proportion of significantly imputable genes (false positives rate) by p-value threshold.
(PDF)

**S2 Table. GTEx version 6 comparisons of single-tissue and multi-tissue imputation models using GEUVADIS LCL RNA-Seq expression as validation.** Counts (B-H counts) are based on Benjamini-Hochberg procedure false discovery rate of 0.05. The last column displays the number of counts at p-value threshold 0.05 (without any corrections).
(PDF)

**S3 Table. GTEx version 7 comparisons of single-tissue and multi-tissue imputation models using GEUVADIS LCL RNA-Seq expression as validation.** Counts (B-H counts) are based on Benjamini-Hochberg procedure false discovery rate of 0.05. The last column displays the number of counts at p-value threshold 0.05 (without any corrections).
(PDF)

**S4 Table. GTEx version 8 comparisons of single-tissue and multi-tissue imputation models using GEUVADIS LCL RNA-Seq expression as validation.** Counts (B-H counts) are based on Benjamini-Hochberg procedure false discovery rate of 0.05. The last column displays the number of counts at p-value threshold 0.05 (without any corrections).
(PDF)

**S5 Table. Comparison of all multi-tissue methods.** We applied SWAM to all combinations of GTEx and DGN resources. For the GTEx resources, we always used every tissue available. In version 6, this comprised of 44 tissues. For version 7, there were 48 tissues and version 8 contained 49 tissues. For the sake of consistency, our target tissue for each of these combinations was GTEx v6 LCL.
(PDF)

**S6 Table. TWAS association signals for SWAM.** We used SWAM to derive an tissue-specific model for every GTEx version 6 tissue, and used these models as inputs to metaXcan to infer TWAS results. As mentioned in the Materials and Methods section, the HDL and LDL traits were from Global Lipids Genetics Consortium (GLGC) and Type-2 Diabetes (T2D) from the DIAGRAM consortium.
(PDF)

**S7 Table. TWAS association signals for UTMOST.** These models were also derived from GTEx version 6 tissues using the UTMOST method. Models were downloaded from *https://*

*github.com/Joker-Jerome/UTMOST*
(PDF)

**S8 Table. TWAS association signals for prediXcan (single-tissue).** TWAS results via metaX-can using prediXcan single tissue models derived from GTEx version 6 tissues
(PDF)

**S9 Table. Genomic inflation factor for every TWAS dataset.** To control for false-positives, we calculated the genomic inflation factor for every TWAS dataset and used these to adjust the significance threshold for each analysis accordingly.
(PDF)

## Author Contributions

**Conceptualization:** Andrew E. Liu, Hyun Min Kang.

**Data curation:** Andrew E. Liu, Hyun Min Kang.

**Formal analysis:** Andrew E. Liu, Hyun Min Kang.

**Funding acquisition:** Hyun Min Kang.

**Investigation:** Andrew E. Liu, Hyun Min Kang.

**Methodology:** Andrew E. Liu, Hyun Min Kang.

**Resources:** Hyun Min Kang.

**Software:** Andrew E. Liu, Hyun Min Kang.

**Supervision:** Hyun Min Kang.

**Validation:** Andrew E. Liu, Hyun Min Kang.

**Visualization:** Andrew E. Liu, Hyun Min Kang.

**Writing – original draft:** Andrew E. Liu.

**Writing – review & editing:** Andrew E. Liu, Hyun Min Kang.

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
