## [Decision Letter · Decision Letter 0]

20 May 2021

Dear Dr Liu,

Thank you very much for submitting your Research Article entitled 'Meta-imputation of transcriptome from genotypes across multiple datasets using summary-level data.' to PLOS Genetics.

The manuscript was fully evaluated at the editorial level and by independent peer reviewers. The reviewers found the work interesting and could be useful for transcriptome-wide association studies. However, they also raised several concerns that need to be addressed. Notably, it is problematic to use the same dataset to learn model weights and test for accuracy. The simulations need to better reflect complexities seen in real data. Also, the software link provided in the manuscript does not work. Based on the reviews, we will not be able to accept this version of the manuscript, but we would be willing to review a much-revised version. We cannot, of course, promise publication at that time.

If you decide to revise the manuscript for further consideration at PLOS Genetics, please aim to resubmit within the next 60 days, unless it will take extra time to address the concerns of the reviewers, in which case we would appreciate an expected resubmission date by email to plosgenetics@plos.org.

[LINK]

We are sorry that we cannot be more positive about your manuscript at this stage. Please do not hesitate to contact us if you have any concerns or questions.

Yours sincerely,

Mingyao Li

Associate Editor

PLOS Genetics

David Balding

Section Editor: Methods

PLOS Genetics

Reviewer's Responses to Questions

**Comments to the Authors:**

Reviewer #1: The authors presents a very interesting and useful approach to aggregate reference data of multiple tissues and cohorts for TWAS. I have the following major comments:

1. The authors use the same data set to learn model weights and test accuracy. The test results will be “over fitted” because the Optimal Model Weights were learned using the same data set. The authors would need to either use another independent data set as test data or split the current test data into validation data for training model weights and an independent test data. This should also be done for all simulation studies.

2. The simulation study should be better designed to mimic real TWAS. Modeling only 1 or 2 true causal eQTL is not realistic for TWAS as the authors can see how many number of SNPs would be tested by PrediXcan using GTEx reference data. Also, the authors only consider independent SNPs in the simulation studies, which is not realistic as the gene expression imputation models are trained using all cis-SNPs that are in LD and are of about thousands per gene.

3. I do not think the authors need to discuss using all versions of GTEx data as the most recent GTEX V8 would contain all samples from the previous versions. I think the authors should focuse on using GTEx V8 data in this paper.

4. The authors should discuss the proportion of overlapped significant TWAS genes by Single Tissue PrediXcan, UTMOST and SWAM method, and the number of genes that only identified by UTMOST or SWAM method, or single tissue PrediXcan.

Reviewer #2: Liu and Kang proposed SWAM (Smartly Weighted Averaging across Multiple datasets) to improve the gene expression imputation of the target tissue using pre-trained imputation models from other relevant tissues. The weight for each tissue depends on the correlation among tissues, the correlation between imputed and observed expression in the testing set, and a penalty tuning parameter. They conducted simulations and real data analyses to demonstrate the advantage over existing methods in imputation accuracy and the number of genes identified in transcriptome-wide association studies (TWAS). Below are some comments that may improve the manuscript.

The title “Meta-imputation of transcriptome from genotypes across multiple datasets using summary-level data” gave the impression that the proposed method only requires summary-level data. However, this is not true according to the following excerpt.

“Using the individual level genotypes and expression of only the reference tissue, SWAM integrates imputation models trained from different tissues and datasets without requiring individual-level data except for the reference tissue.”

So “using summary-level data” is a little bit misleading. The proposed method SWAM does require the individual-level expression of the reference tissue to calculate weights to impute gene expression in the target tissue. Only if the authors can upload trained weights in a way similar to predictDB or FUSION, the method would be fully based on summary-level data. Otherwise, the authors may need to clarify this potential misunderstanding.

SAWM relies on a tuning parameter lambda to obtain stable meta weights. Right now, it was empirically set as 3. Can the tuning parameter lambda be selected via cross-validation?

It is good that SWAM identified more genes in TWAS. But it may be necessary to demonstrate that SWAM controls for false positives, either through simulations or the histogram/QQ plot of the p-values.

UTMOST was compared in the real data application. Why was it not compared in the simulations?

The numbering of figures in the main text and supplementary and tables is incorrect and confusing.

Fig 6D (should be Fig 2D) was cited for sample size, but it is actually about heritability, while 2F is about sample size. Both Figs 2E and 2F were not cited. The caption of subfigures of Fig 2 ABC was mismatched with the figures.

Line 292, Fig S3 should be cited other than S2.

“The software package for SWAM is available at https://github.com/aeyliu/swam, including the datasets used in this manuscript.”

The link does not work.

Some typos:

Line 160: to maximize expected mean squared error

Line 481, “each simulation was repeated for 1,000 times”, while in the caption of Fig 2, it said, “we run each simulation 10,000 times”. Which one is correct?

Line 793, the second equation contradicts with that in line 797

Line 115, “MultiXcan, *does not impute transcriptomes*, but performs a multi-tissue TWAS across all tissues by including each tissue-specific *imputed expression* as a predictor variable”

The bold parts contradict each other. It requires a clearer description.

The references wrongly put consortiums as the first author when it is actually listed in the middle. This problem applies to multiple cited references.

Some grammar errors:

Line 113, transcriptome imputation model A simultaneously across all tissues

Line 193, the power of SWAM and naïve average keepS increasing

Line 923, (B) depicts are scenario

Line 944, Each cell in this heatmap depict

**Have all data underlying the figures and results presented in the manuscript been provided?**

Reviewer #1: Yes

Reviewer #2: None

PLOS authors have the option to publish the peer review history of their article (what does this mean?). If published, this will include your full peer review and any attached files.

Reviewer #1: No

Reviewer #2: No

---

## [Decision Letter · Decision Letter 1]

5 Sep 2021

Dear Dr Liu,

Thank you very much for submitting your Research Article entitled 'Meta-imputation of transcriptome from genotypes across multiple datasets by leveraging publicly available summary-level data' to PLOS Genetics.

The manuscript was evaluated at the editorial level and by independent peer reviewers. Although many issues have been adequately addressed in the revision, Reviewer 1 still wants to see a detailed comparison with UTMOST in simulations and is not satisfied with your explanations of why this is difficult.  The editors tend to agree with the reviewer.  In particular, in the analysis of real data the genomic inflation factor of SWAM is 1.15 and so it is necessary to use simulations to understand how well SWAM can control false positives and, since UTMOST is the main competitor, it is necessary to compare its false positive rate control with UTMOST. Despite this being difficult, it is necessary that you either find a way to do a fair comparison (usually the authors will be helpful when asked for advice on how to implement their method) or else provide a convincing demonstration that UTMOST cannot adequately address this problem.

We hope you will be able to address this one remaining issue and we look forward to receiving your further-revised manuscript.  Please include a description of the changes you have made in the manuscript.

If you decide to revise the manuscript for further consideration at PLOS Genetics, please aim to resubmit within the next 60 days, unless it will take extra time to address the concerns of the reviewers, in which case we would appreciate an expected resubmission date by email to plosgenetics@plos.org.

[LINK]

We are sorry that we cannot be more positive about your manuscript at this stage. Please do not hesitate to contact us if you have any concerns or questions.

Yours sincerely,

Mingyao Li

Associate Editor

PLOS Genetics

David Balding

Section Editor: Methods

PLOS Genetics

Reviewer's Responses to Questions

**Comments to the Authors:**

Reviewer #1: The authors have addressed most of my comments. I still think the authors shall compare with UTMOST in their simulations and real data with GTEx V8 data. As suggested by the other reviewer, the authors shall evaluate Type I Error in simulation studies instead of just showing the genetic control factor lambda values, and compare with UTMOST in simulation studies.

Reviewer #2: My comments have been addressed.

**Have all data underlying the figures and results presented in the manuscript been provided?**

Reviewer #1: Yes

Reviewer #2: None

PLOS authors have the option to publish the peer review history of their article (what does this mean?). If published, this will include your full peer review and any attached files.

Reviewer #1: No

Reviewer #2: No

---

## [Decision Letter · Decision Letter 2]

7 Jan 2022

Dear Dr Liu,

We are pleased to inform you that your manuscript entitled "Meta-imputation of transcriptome from genotypes across multiple datasets by leveraging publicly available summary-level data" has been editorially accepted for publication in PLOS Genetics. Congratulations!  Reviewer 2 has some minor comments that we ask you to address briefly in preparing your final version for publication.

Yours sincerely,

Mingyao Li

Associate Editor

PLOS Genetics

David Balding

Section Editor: Methods

PLOS Genetics

Comments from the reviewers:

Reviewer #1: My comments have been well addressed.

Reviewer #2: After reading the revision, I have the following minor comments:

1. Fig 1 does not completely reflect the idea of SWAM. Step 2, weight w_j ~ cov(Y_T, \\hat{Y}_j), where j denotes the jth model, Y_T is the gene expression matrix of the target tissue, and \\hat{Y}_j is the predicted gene expression using the jth model. However, in addition to this covariance, the weight in the Methods section is described to also depend on the other prediction models, not only model j.

2. The authors may be more careful with the notations. In Fig 1, the authors used T to denote the target tissue but used little t in line 500? For "mt" in line 894, does it mean multiple tissues?

**Have all data underlying the figures and results presented in the manuscript been provided?**

Reviewer #1: Yes

Reviewer #2: None

PLOS authors have the option to publish the peer review history of their article (what does this mean?). If published, this will include your full peer review and any attached files.

Reviewer #1: No

Reviewer #2: No

**Data Deposition**

http://datadryad.org/submit?journalID=pgenetics&manu=PGENETICS-D-21-00570R2

**Press Queries**

---

## [Editor Report · Acceptance letter]

25 Jan 2022

PGENETICS-D-21-00570R2 

Meta-imputation of transcriptome from genotypes across multiple datasets by leveraging publicly available summary-level data 

Dear Dr Liu, 

We are pleased to inform you that your manuscript entitled "Meta-imputation of transcriptome from genotypes across multiple datasets by leveraging publicly available summary-level data" has been formally accepted for publication in PLOS Genetics! Your manuscript is now with our production department and you will be notified of the publication date in due course.

With kind regards,

Orsolya Voros

PLOS Genetics

On behalf of:
